# Effects of Contagious Respiratory Pathogens on Breath Biomarkers

**DOI:** 10.3390/antiox13020172

**Published:** 2024-01-29

**Authors:** Nele Kemnitz, Patricia Fuchs, Rasmus Remy, Leo Ruehrmund, Julia Bartels, Ann-Christin Klemenz, Phillip Trefz, Wolfram Miekisch, Jochen K. Schubert, Pritam Sukul

**Affiliations:** Rostock Medical Breath Research Analytics and Technologies (ROMBAT), Department of Anaesthesiology, Intensive Care Medicine and Pain Therapy, University Medicine Rostock, 18057 Rostock, Germany

**Keywords:** metabolic profiling, respiratory virus, pulmonary bacteria, breathomics, volatile biomarkers, real-time mass spectrometry

## Abstract

Due to their immediate exhalation after generation at the cellular/microbiome levels, exhaled volatile organic compounds (VOCs) may provide real-time information on pathophysiological mechanisms and the host response to infection. In recent years, the metabolic profiling of the most frequent respiratory infections has gained interest as it holds potential for the early, non-invasive detection of pathogens and the monitoring of disease progression and the response to therapy. Using previously unpublished data, randomly selected individuals from a COVID-19 test center were included in the study. Based on multiplex PCR results (non-SARS-CoV-2 respiratory pathogens), the breath profiles of 479 subjects with the presence or absence of flu-like symptoms were obtained using proton-transfer-reaction time-of-flight mass spectrometry. Among 223 individuals, one respiratory pathogen was detected in 171 cases, and more than one pathogen in 52 cases. A total of 256 subjects had negative PCR test results and had no symptoms. The exhaled VOC profiles were affected by the presence of *Haemophilus influenzae*, *Streptococcus pneumoniae,* and Rhinovirus. The endogenous ketone, short-chain fatty acid, organosulfur, aldehyde, and terpene concentrations changed, but only a few compounds exhibited concentration changes above inter-individual physiological variations. Based on the VOC origins, the observed concentration changes may be attributed to oxidative stress and antioxidative defense, energy metabolism, systemic microbial immune homeostasis, and inflammation. In contrast to previous studies with pre-selected patient groups, the results of this study demonstrate the broad inter-individual variations in VOC profiles in real-life screening conditions. As no unique infection markers exist, only concentration changes clearly above the mentioned variations can be regarded as indicative of infection or colonization.

## 1. Introduction

Since the dawn of the present pandemic, public health resources, clinical focus, and translational research efforts towards respiratory infections and coinfections have been employed globally more than ever before. Certain long-existing pathogens, e.g., *Streptococcus pneumoniae* (*S. pneumoniae*), *Haemophilus influenzae (H. influenzae)*, and Rhinovirus, are largely associated with the most frequent respiratory infections [1,2], where the disease symptoms are similar to those of SARS-CoV-2 infection. Despite state-of-the-art medical knowledge, biomedical facilities, and technological advancements, the differential diagnosis, monitoring, and management of highly contagious respiratory pathogens face daily obstacles due to the rising complexity of the disease mechanisms, manifestations, and host responses [3,4,5]. While rapid point-of-care diagnostics are largely developed and optimized for population screening, the continuous/repeated monitoring of disease progression and the host’s response to the pathogen and/or therapy still remains challenging [6,7,8].

Besides in vitro and/or in vivo pre-clinical (animal model) studies [9,10,11], clinical investigations are extremely important to mechanistically address complex disease mechanisms and host–(microbiome)–pathogen interactions [12,13,14]. Thus, the profiling of pathobiology-driven in vivo metabolic/biochemical changes at the host cellular and systemic microbial levels may address the primary challenges in human medicine and pave the way for new prophylactic and/or therapeutic targets. Volatile organic compounds (VOCs) of various endogenous and/or exogenous origins and chemical classes are continuously exhaled in trace concentrations (the ppbV–pptV range) [15]. Apart from physiological [16,17,18,19,20], metabolic [21,22,23,24,25,26], and therapeutic monitoring [27,28], the real-time mass-spectrometry-based profiling of quantified differences/changes in the alveolar concentrations of breath VOCs could provide unique insights into the disease mechanisms of respiratory infections and the host response [13,29]. Recently, we have developed an advanced setup for the safe breath sampling and VOC-based monitoring of patients experiencing highly contagious respiratory infections [30] and have successfully applied it to the breath screening of hundreds of subjects at a COVID-19 test center, as well as monitoring critically ill (mechanically ventilated) patients at an intensive care unit.

Most recent cross-sectional breathomics studies have reported differences in VOC profiles between pre-selected SARS-CoV-2 patient groups and healthy volunteers and proposed unique/specific diagnostic markers/patterns/signatures for disease detection [29,31,32]. Our previous study tried to create a real-life screening scenario in a COVID-19 test center [13] but could not confirm the previously postulated specific VOC marker sets. However, we found differences probably related to the host response in COVID-19 patients. As the host response may not be highly specific to the causative pathogen, the effects of other common respiratory pathogens on breath biomarker profiles have to be explored in more detail.

Here, we analyzed the exhaled VOC profiles of patients who tested positive for the most common pulmonary mono-pathogens (viral and bacterial) and co-pathogens (viral–bacterial and bacterial–bacterial) within the screened cohort.

## 2. Methods

### 2.1. Experimental Setup, Ethics, and Human Subjects

Advanced experimental setups for breath sampling and patient monitoring were developed in compliance with the unavoidable infection safety mandates of the present global pandemic [30]. Setups and associated methods were successfully applied at our university’s COVID-19 test center without interfering with routine screening activities [13].

Upon approval by the institutional ethics committee (IEC) of the University Medicine Rostock (approval No. A 2020 0085), a single-center prospective observational study was conducted on 708 consecutively recruited and non-pre-selected (i.e., randomly selected to avoid selection bias) subjects as per the amended ‘Declaration of Helsinki’ guidelines. The initial setup was created to collect breath biomarker profiles from spontaneously breathing human subjects using a proton transfer reaction—time of flight—mass spectrometer (PTR-ToF-MS) in parallel with real-time quantitative polymerase chain reaction (RT-qPCR) (for SARS-CoV-2 pathogen) and multiplex PCR (for other respiratory pathogens) analyses. While the data on SARS-CoV-2 infection from this cohort were published earlier [13], within this study, we explored the effects of common respiratory pathogens (e.g., *S*. *pneumoniae, H*. *influenzae,* and Rhinovirus) on exhaled VOC profiles.

Amongst the 708 initially recruited subjects, 36 patients were SARS-CoV-2-positive, and 193 had symptoms without positive test results. These patients were excluded from the analysis within this study. Amongst the remaining 479 subjects, 256 subjects had negative tests without any symptoms, and 223 patients tested positive for respiratory pathogen(s) with and without flu-like symptoms. The demographic data of subjects are listed in Table 1.

In order to avoid pre-selection bias and to create real-life screening conditions, we conducted all measurements in a prospective screening scenario—without considering extrinsic factors, e.g., subjects’ lifestyles, habits, habitats, chronic comorbidities, and diet/therapy.

### 2.2. Inclusion and Exclusion Criteria

All subjects above the age of 14 were included in our study. Participation was voluntary and was not a mandatory requirement at the test center.

Subjects less than 14 years of age were excluded from the study. Unwillingness to provide demographic information and refusal to sign the informed consent form were also considered as exclusion criteria.

### 2.3. Multiplex PCR Test for Respiratory Pathogens

Routinely, one throat swab sample was taken from each participant in the COVID-19 screening center. A commercially available standardized multiplex RT-qPCR test (Allplex^TM^, Resp. Panel 1A,2,3,4—Seegene) was performed for the detection of respiratory bacteria and viruses. A species spectrum of seven bacteria and 19 viruses was detectable by means of this system.

### 2.4. Reporting of Disease Symptoms

The presence or absence of classical flu-like symptoms (cough, cold, fever, headache, runny nose, difficulty breathing, throat pain, loss of smell and/or taste, abdominal pain, pneumonia) was assessed by means of a questionnaire and patients were classified as symptomatic or asymptomatic.

### 2.5. Infection Safety Measures and Breath Sampling Protocol

The customized mouthpiece attachments incorporating mainstream and side-stream filters and the detailed analytical methods have been described in our previous study [30]. In brief, the sampling area for participants or patients was separated from that of the investigator(s). Only the silico-steel transfer line (6 m, long and heated at 100 °C) of the PTR-ToF-MS entered the sampling area and was attached to the mouthpiece via the side-stream filter. The sampling area was disinfected and ventilated well after each examination, and all contaminated materials were disposed of after use. For more details, see the Methods section of our recent clinical study [13].

Subjects rested for 5 min in a sitting posture [19] and then removed their face mask to perform normal oral breathing through the sterile mouthpiece for 3 min in accordance with our state-of-the-art breathing protocol [17]. We used both exhaled alveolar and inspiratory room air samples for breath VOC analysis.

### 2.6. PTR-ToF-MS Measurements of Breath Composition and VOC Data Processing

A PTR-ToF-MS 1000 (Ionicon Analytik GmbH, Innsbruck, Austria) with a mass resolution of 1000–2000 m/Δm was applied for continuous side-stream sampling and the breath-resolved analysis of VOCs under pre-optimized conditions [30]. The PTR sampling flow (100 mL/min), time resolution (200 ms), drift tube temperature (75 °C), voltage (610 V), and pressure (2.3 mbar) were adjusted to attain the desired E/N ratio of 139 Td. After automatically recording one data file/min, the mass scale was recalibrated upon the defined masses *m/z* 21.022 (H_3_O^+^-isotope), *m/z* 29.998 (NO^+^), and *m/z* 59.049 (protonated C_3_H_6_O).

The PTR-MS viewer software (version 3.228) was used for raw data processing. Continuously measured VOC data (in counts per seconds) were normalized to primary ion (H_3_O^+^) counts and were assigned to expiratory (end-tidal) and inspiratory (room air) phases via the custom-made ‘breath tracker’ algorithm, which uses endogenous acetone as a tracker mass [30,33]. Every 15 min, the room air was analyzed. Room air VOC concentrations were used to detect the acute effects of ambient/exogenous contamination on the exhaled VOC profiles. Substances with higher concentrations in the room air than in alveolar air were excluded from further analysis. For more details, see the Methods section of our recent clinical study [13].

### 2.7. Quantification of VOCs

In total, 44 VOCs were quantified. VOCs of potentially exogenous origin and those emitted from the protective viral filters were excluded (Appendix A) from further data analysis [30]. Exogenous VOCs are typical compounds (e.g., formaldehyde, formic acid, isopropanol, benzene, toluene, acetonitrile, acrolein, furan, cyclopentanone, crotonic acid, monoterpenes, etc.) well known to appear in breath from the environment, disinfectants, lifestyle habits, diet, preexposure, etc. VOCs with concentrations below the limit of detection (LOD, i.e., mean of blank +3 SD) were excluded from statistical analysis. The quantification of VOCs was partly performed via reaction rate coefficients (*k*-rates) between the VOC and H_3_O^+^ (at the E/N ratio of 140 Td) [13,34] and partly via a multi-component mixture of matrix-adapted standard reference substances by means of a liquid calibration unit (LCU, Ionicon Analytik GmbH, Innsbruck, Austria) [13,35]. For LCU-based calibrations, VOCs were analyzed in different concentrations and with different sample humidity levels. The influence of humidity on the measured VOC intensities, calibrations, and LOD/LOQ was systematically evaluated. VOCs with concentrations below the LOD were excluded from statistical analysis. For more details, see the Methods section of our recent clinical study [13].

### 2.8. Statistical Analysis

In each participant, VOC concentrations were averaged over one minute of measurement. Statistical analysis was performed on VOC data from the 2nd minute of all measurements. As the VOC data were not normally distributed, exhaled VOC concentrations in different groups were compared by means of the Kruskal–Wallis ANOVA on ranks (Sigma Plot v. 14, SYSTAT, Frankfurt, Germany); a *p*-value ≤ 0.05 was regarded as statistically significant. In order to control for type I and type II errors, significance values were adjusted by means of a Bonferroni correction for multiple comparisons and a high test power of 0.95 was considered, respectively. Table 2 shows the performed statistical queries (Q1–Q13).

### 2.9. Q1–Q6 (All Mono-Pathogens)

VOC profiles from *H. influenzae*-positive (*n* = 97), *S. pneumoniae*-positive (*n* = 40), Rhinovirus-positive (*n* = 34), and healthy participants (*n* = 256) were compared to each other.

### 2.10. Q7–Q8 (Asymptomatic Mono-Pathogens)

VOC profiles from the asymptomatic positive subgroups of *H. influenzae* (*n* = 55), *S. pneumoniae* (*n* = 25), and the healthy cohort (*n* = 256) were compared to each other.

### 2.11. Q9–Q10 (Symptomatic Mono-Pathogens)

VOC profiles from the symptomatic positive subgroups of *H. influenzae* (*n* = 42), *S. pneumoniae* (*n* = 15), and the healthy cohort (*n* = 256) were compared to each other.

### 2.12. Q11–Q13 (Co-Pathogens)

VOC profiles from participants experiencing (bacterial–bacterial or viral–bacterial) co-infections with *H. influenzae* + *S. pneumoniae* (*n* = 24), *H. influenzae* + Rhinovirus (*n* = 16), and *S. pneumoniae* + Rhinovirus (*n* = 12) were compared with those of the healthy cohort (*n* = 256).

In order to observe the inter-individual physio-metabolic variations under the presence of pathogens and symptoms, coefficients of variation (RSD in %) in the different study groups were calculated by rating the sample standard deviation over the sample mean.

In order to understand the correlations between differentially expressed endogenous VOCs, a dimension reduction factor analysis (factor extraction via principal component method, factor scores via regression method, and 1-tailed significance at *p*-value ≤ 0.05, regression values R ≥ 0.500) was performed within each study group and subgroup in SPSS (v. 27).

## 3. Results

Out of 223 patients with respiratory pathogens, 171 had only one pathogen, 52 had co-pathogens with or without flu-like symptoms, and 256 subjects had no respiratory pathogens or symptoms. In the group with mono-pathogens, 97 participants tested positive for *H. influenzae*, 40 for *S. pneumoniae*, and 34 for Rhinovirus. In the group with co- pathogens, 24 participants tested positive for *H. influenzae* + *S. pneumoniae*, 16 for *H. influenzae* + Rhinovirus, and 12 for *S. pneumoniae* + Rhinovirus. Demographic data from all subjects are shown in Table 1.

A heat map of the relative differences in the exhaled alveolar VOC profiles from the different study groups is presented in Figure 1. The heat map represents the non-quantitative expression of the normalized (onto maximum) mean values of the VOC concentrations in seven groups and subgroups based on the presence of mono-pathogens, co-pathogens, and symptoms. The groups consisted of healthy subjects without any pathogens or symptoms; all cases positive for *H. influenzae* only, *S. pneumoniae* only, and Rhinovirus only; and all cases positive for *H. influenza* + *S. pneumoniae*, *H. influenzae* + Rhinovirus, and *S. pneumoniae* + Rhinovirus, respectively. The selection criteria for VOCs are described in the Methods section. *H. influenzae*- and *S. pneumoniae*-positive cases were sub-grouped based on the presence and absence of disease symptoms.

The distribution of the non-quantitative expression of VOCs shown in Figure 1 is presented through violin plots in Appendix A.

Quantified median values and percentiles for all 44 substances are presented in Appendix A. VOCs of potentially exogenous origin and those emitted from the protective viral filters [30] were excluded (Appendix A) from further data analysis.

Statistical comparisons based on the exhaled alveolar VOC concentration between study groups and subgroups are presented in Table 2 as queries Q1–Q13. Out of 44 volatile substances, acetone, acetic acid, dimethyl sulfide, pentanal, and limonene were expressed at varying levels, as evidenced by pairwise multiple comparisons. Statistically significant differences in substance concentrations (*p*-value ≤ 0.05) are marked in bold. The instrumental LOD and LOQ values for these VOCs are presented in Appendix A.

In accordance with Table 2, boxplots are presented in Figure 2 that demonstrate the distributions of these five VOCs in the different study groups and subgroups. Statistically significant differences (*p*-value ≤ 0.05) are marked with ‘#’.

The coefficients of variation or relative standard deviations (RSDs in %) of the five VOCs within the study groups and subgroups are presented in Table 3. Pairwise correlations between these five VOCs within all study groups and subgroups are presented in Table 4 along with their statistical significance. Statistically significant regressions (one-tailed significance at *p*-value ≤ 0.05, R-value ≥ 0.500) are marked in bold.

## 4. Discussion

In contrast to previous studies with pre-selected patient groups, the results of this study demonstrate the broad inter-individual variations in VOC profiles in a real-life screening scenario. As no unique pathogen biomarkers exist, only concentration changes clearly above the mentioned variations can be regarded as indicative for bacterial colonization or infection.

Depending on the presence or absence of pathogens and symptoms, we observed differences in exhaled alveolar VOC concentrations. Based on the VOCs’ origins, the observed concentration changes may be attributed to energy metabolism (acetone), systemic microbial immune homeostasis (dimethyl sulfide, acetic acid), oxidative stress (pentanal), and antioxidative defense (limonene).

***Effects on oxidative stress and antioxidative defense*:** Pathogens primarily down-regulate the host’s antioxidative defense in order to facilitate pathogenesis [36,37]. This further elevates reactive oxygen species (e.g., hydroxyl radical) production, leading to oxidative stress. Reactive aliphatic aldehydes (e.g., α, β-unsaturated and saturated) are thus produced due to lipid peroxidation [38].

Significantly increased pentanal exhalation in the asymptomatic cases with *S. pneumoniae*, therefore, indicates increased redox reactions at the cellular level [39]. *S. pneumoniae* produces hydrogen peroxide and pneumococcal autolysin [40] as metabolic byproducts, which contribute to its virulence. The host’s oxidative stress response is stimulated through airway inflammation, and oxidative resistance in broncho-pulmonary epithelial cells is modulated [41]. Thus, increased host airway–lung epithelial oxidative stress elevates pentanal production significantly in the presence of *S. pneumoniae*, especially in asymptomatic subjects.

Rhinovirus is known to downregulate mitochondrial respiration within human bronchial epithelial cells and to increase proton leakage in vitro [42]. As a consequence, pro-inflammatory cytokines (e.g., interleukin-8) are released [43], which regulate oxidative stress (mitochondrial/endoplasmic reticular) exposure in order to stabilize mitochondrial function [44]. In patients experiencing co-infection with rhinovirus and *S. pneumoniae*, exhaled pentanal did not increase due to immune-mediated competition between these pathogens.

Limonene and its metabolites act as cellular antioxidants by removing free radicals [45]. The hepatic microsomal metabolism of limonene boosts the antioxidative defense and anti-inflammatory response [46,47]. Increased limonene exhalation in the presence of Rhinovirus mirrors the anti-inflammatory host response against the viral oxidative stress and injuries in the respiratory tract.

***Effects on host’s energy metabolism*:** Acetone mainly originates from glycolysis and lipolysis [48]. Gram-negative (e.g., *H. influenzae)* and Gram-positive bacterial pathogens like *S. pneumoniae* primarily use carbohydrates for energy production [49,50]. As the energy demand is high for colonization, invasion, and infection, acetone exhalation increased in comparison to the healthy cohort [51,52] when bacteria were present. In contrast, Rhinovirus does not have its own metabolism [53,54] and is known to reduce glycolysis [55]. Consequently, acetone exhalation remained unaffected whenever Rhinovirus was present.

In the case of the co-existence of *H. influenzae* and *S. pneumoniae*, one may assume that acetone exhalation would increase due to cumulative effects. A recent study [56] demonstrated the synergy between *H. influenzae* and *S. pneumoniae*. Both species reached higher cell densities in the coculture compared with the monoculture. In a (biofilm) coculture, the physical contact of *H. influenzae* with *S. pneumoniae* resulted in the increased expression of virulence factors by *H. influenzae*.

In contrast, here, breath acetone did not change, which indicates immune-mediated competition between these two bacteria in vivo [57]. It is noteworthy that the spxB of *S. pneumoniae* is responsible for hydrogen peroxide production, and the above study [56] showed that it was upregulated in the coculture biofilm of *S. pneumoniae* and *H. influenzae*. Studies have shown that hydrogen peroxide produced by *S. pneumoniae* kills *H. influenzae* and other pathogens in vitro [58]. Neuraminidase produced by *S. pneumoniae* desialylates sialic acid—a lipo-oligosaccharide structure of *H. influenzae*—possibly reducing bacterial viability in vivo [59,60].

***Effects on systemic microbial metabolism, immunomodulatory, and inflammatory response*:** Short-chain fatty acids (SCFAs) such as acetic acid and organosulfurs such as dimethyl sulfide are largely produced in the lower gut through dietary fiber/starch fermentation [61,62] and the methylation of bacteria [63,64]. These bacteria maintain the anaerobic conditions to regulate intestinal permeability, nutrient absorption, and the adaptive immune response [65,66,67]. Most importantly, the bio-chemical interplay between the intestinal and pulmonary microbiota regulates upper and lower respiratory tract health, preventing infections via the ‘gut–lung axis’ [68,69,70].

Pre-clinical and clinical models have demonstrated that the SCFA acetate is produced by the pulmonary and gut microbiome as an immunomodulatory response against respiratory pathogens [71,72,73,74]. Thus, increased breath acetic acid in the cohorts with only one pathogen indicates a host–microbial response against pathogens.

There is evidence that the gut microbiota contributes to the host’s methionine (i.e., precursor of S-adenosylmethionine) metabolism and, thereby, regulates the macrophage-mediated inflammatory response [75,76,77]. Thus, the high dimethyl sulfide concentrations in the asymptomatic cases of bacterial presence (significant for *S. pneumoniae*) and decreased concentrations in the presence of disease symptoms (significant for *H. influenzae*) indicate a change in the corresponding microbial activity between early and progressive bacterial pathogenesis. As human Rhinovirus remains restricted to the respiratory tract epithelium, nose, and nasopharynx and does not manifest elsewhere (including gut), no effects are seen on organosulfur exhalation.

Surprisingly, none of the co-pathogens altered the exhaled SCFA or organosulfur profiles. This is most likely due to the cumulative dysbiosis of the systemic microbiome [78,79]—as observed in the case of viral–bacterial superinfection and secondary bacterial pneumonia.

It is noteworthy that all these results are based on single point measurements and, therefore, will have considerable intra-individual variations if measured repeatedly/longitudinally. Although we may certainly generate ROC curves by applying differential features (e.g., from violin plots or boxplots) and may calculate certain test sensitivities and specificities to identify differential diagnostic markers for these respiratory pathogens, we restrained ourselves from such practice. The refitting of pre-sorted/pre-processed data to the present model will undoubtedly result in high sensitivity and specificity but will have no clinical/real-life relevance without considering enough independent measures.

When considering another population/ethnicity, our present observations may vary based on genetic predispositions, infection rates, population immunity regimes, bacterial strains, viral variants, and comorbidities. Pathogen loads were not measured within the present study as the presence or absence of pathogens was confirmed via RT-PCR tests. Similarly, inflammatory blood markers were not considered within the present observational design in order to avoid any invasive intervention that was not part of the routine tests conducted at the COVID-19 screening center. Therefore, the assignment of bacterial colonization, proliferation, the host’s inflammatory response, and immune-mediated inter-pathogenic competition relies upon the presence or absence of disease symptoms and available knowledge from in vitro and in vivo studies. Despite the relatively small sample size in the groups and subgroups, we could attain a test power of 0.90 at an alpha value of 0.05 in order to detect up to 5% differences in exhaled VOC trace concentrations. As indicated above, discrimination between pathogen colonization and infection was beyond the scope of the present study. Extrinsic and intrinsic confounders due to lifestyle [80,81,82], comorbidities, and treatment, etc., may influence VOC exhalation. In order to avoid any pre-selection bias in our study design and to reflect real-life screening conditions, we had to disregard potential physio-metabolic confounders, e.g., nutrition, endocrine milieu, lifestyle habits (e.g., exercise), or hormonal status, and any effects of comorbidities and/or concurrent medication/therapy. While acute respiratory and hemodynamic effects were minimized by our state-of-the-art sampling procedures, certain metabolic effects (such as nutrition or hormonal changes) on VOC profiles will always co-exist in clinical screening studies.

Therefore, any potential (diagnostic) claims should consider this dynamic behavior of VOC profiles. Only if the acute effects of the pathophysiological impact (such as bacterial presence) on VOC profiles are more pronounced than this regular physiological bias can VOC tests be proposed for broad-scale screening tests.

Based on VOCs’ origins, the observed concentration changes may be attributed to energy metabolism (acetone, acetic acid), systemic microbial immune homeostasis (dimethyl sulfide, acetic acid), oxidative stress (pentanal), and antioxidative defense (limonene). Relatively large variations in symptomatic cohorts indicate a broad inter-individual range of host–microbial responses.

As viruses do not have their own metabolism, more complex metabolic interactions and effects were observed in the presence of bacterial pathogens. VOC expression under the presence of co-pathogens indicated immune-mediated competition.

Endogenous VOCs did not differ significantly between patients with one pathogen (Q4–Q6) but they differed when compared to the healthy cohort (Q1–Q3). This indicates similar host–microbial responses against these pathogens. This was also reflected within the group-wise distribution of inter-individual variations (RSDs). Correlations between VOCs of different endogenous origin suggest unknown drivers of pathogenesis that require further longitudinal investigations. Our findings indicate the potential of VOC profiling in monitoring the progression of respiratory infections and responses to administered therapy, and to elucidate new antibiotic/antiviral targets. A recent review has discussed the importance of genomic information in understanding the pathobiology of respiratory viral infections and to plan strategic managements [83]. Similarly, our recent multi-omics investigation has also discovered the actual endogenous origin and metabolic pathway of a breath VOC biomarker (isoprene) in humans [26]. Thus, cross-omics investigations of downstream effects under the presence of respiratory pathogens may elucidate unexplored in vivo metabolic links to exhaled VOCs.

## 5. Conclusions

Depending on the presence of respiratory pathogens and symptoms, we found differences in exhaled alveolar VOC profiles. As VOC expression may non-invasively indicate the promotion or suppression of certain metabolic processes as well as pathogen–microbiome interactions, the findings could be translated to monitor disease progression. Knowledge of ‘host–microbiome–pathogen’ interactions may enhance our present understanding of various pathobiological events, disease manifestation, and the host’s response to infection. As unique or specific infection markers do not exist, only concentration changes above the described variations can be regarded as indicative of bacterial colonization or infection. Due to the highly dynamic nature of exhaled VOCs, the repeated or continuous monitoring of exhaled VOC profiles during infection could provide new insights into host–pathogen interactions. The investigation of substances generated through bacteria can provide unique information on the virus–microbiome interplay.

## Figures and Tables

**Figure 1 antioxidants-13-00172-f001:**
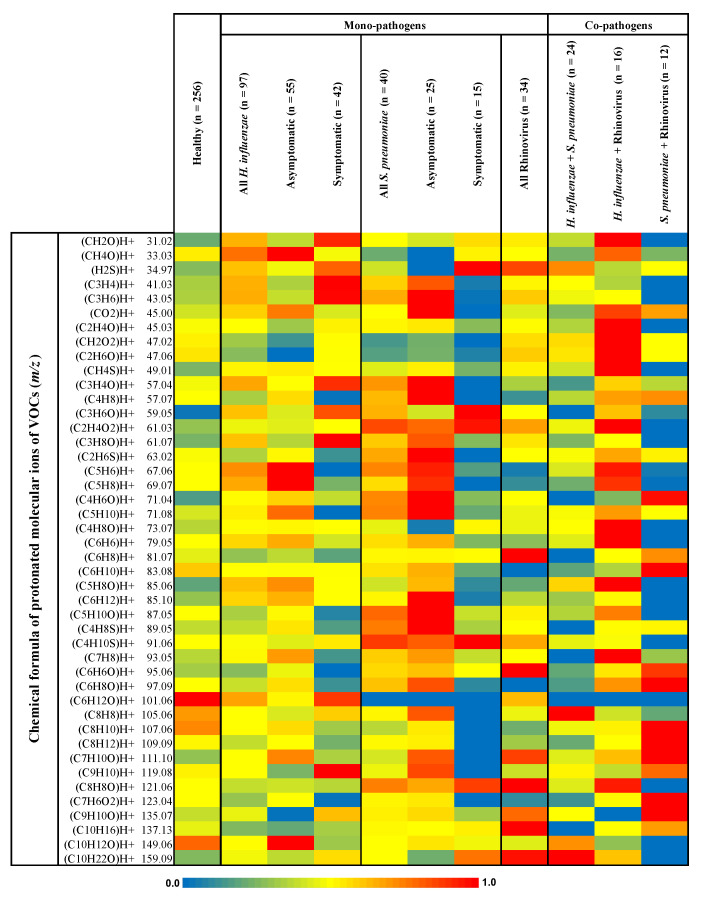
Heat map showing relative differences in exhaled alveolar concentrations between groups. The selection criteria for VOCs are described in the Methods section. Relative differences were observed within and/or between different groups. As denoted through the numeric range between 0.0 and 1.0 within the color scales, red and blue colors indicate relatively high and low values, respectively. The heat map represents the group-wise mean of normalized (to corresponding maximum) concentrations of VOCs in seven groups (with subgroups based on disease symptoms) viz. healthy subjects (without any pathogens or symptoms); all cases positive for *Haemophilus influenzae* only, *Streptococcus pneumoniae* only, and Rhinovirus only; and all cases positive for *H. influenzae* + *S. pneumoniae*, *H. influenzae* + Rhinovirus, and *S. pneumoniae* + Rhinovirus, respectively. *H. influenzae*- and *S. pneumoniae*-positive cases are subgrouped based on the presence and absence of disease symptoms. The numbers of subjects (=*n*) in groups and subgroups are placed as numerical values. The *x*-axis represents the groups and the *y*-axis represents normalized concentrations of protonated VOCs (denominated by chemical formulas and molecular ions).

**Figure 2 antioxidants-13-00172-f002:**
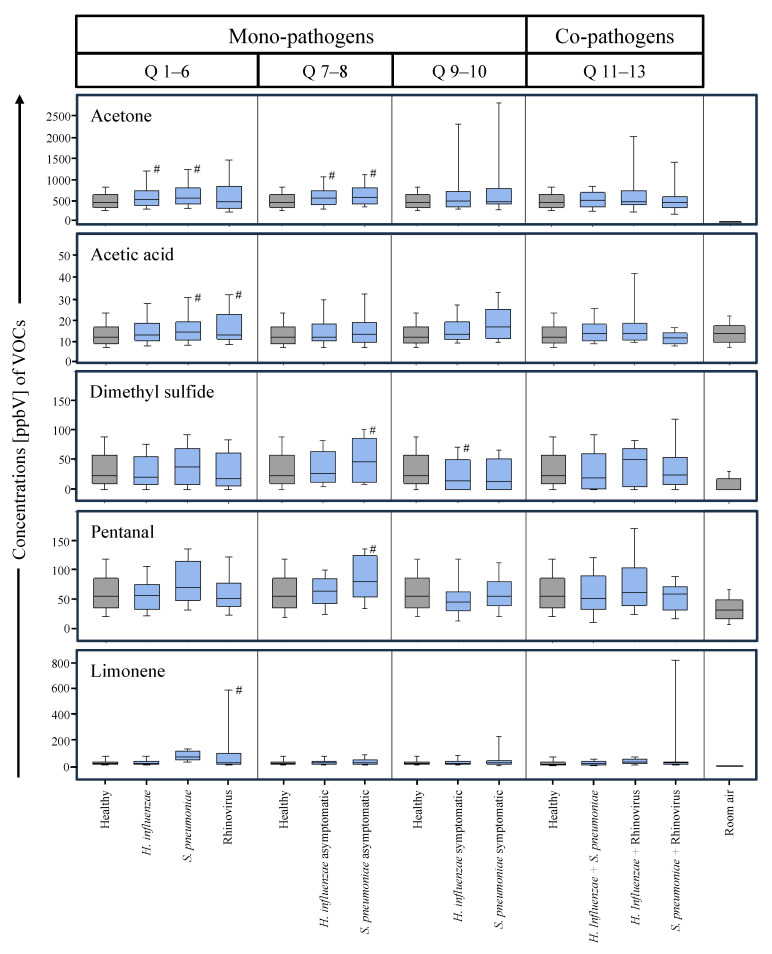
Comparisons of differences in VOC concentrations referring to queries Q1–Q13 of Table 2. Boxplots show exhaled alveolar VOC concentrations from different groups, subgroups, and corresponding room air. The *x*-axis represents the groups and the *y*-axis shows exhaled alveolar concentrations in ppbV. Statistical significances between groups and subgroups were tested by means of Kruskal–Wallis ANOVA on ranks with Bonferroni correction for pairwise multiple comparisons (*p*-value ≤ 0.05). Significant differences with respect to the healthy cohort are indicated by ‘#’. There were no statistical comparisons with room air concentrations.

**Table 1 antioxidants-13-00172-t001:** Demographic data.

	All	Men	Women
N° of subjects (%)	479	249 (52.0)	230 (48.0)
Age [years] (mean ± SD)	39.1 ± 14.2	40.3 ± 13.8	37.9 ± 14.6
N° of respiratory pathogen-positive tested patients (%)	223	115 (51.6)	108 (48.4)
Age of respiratory pathogen-positive tested patients [years] (mean ± SD)	37.3 ± 12.4	38.3± 12.0	36.1 ± 12.8
N° of healthy volunteers (%)	256	134 (52.3)	122 (47.7)
Age of healthy volunteers [years] (mean ± SD)	40.7 ± 15.5	41.9± 15.0	39.5 ± 15.9

**Table 2 antioxidants-13-00172-t002:** Results from inter-group statistical comparisons.

	Mono-Pathogens										Co-Pathogens		
	All Positive Cases						Asymptomatic		Symptomatic		All Positive Cases		
							Positive Cases		Positive Cases				
	Q1	Q2	Q3	Q4	Q5	Q6	Q7	Q8	Q9	Q10	Q11	Q12	Q13
	97 vs. 256	40 vs. 256	34 vs. 256	97 vs. 34	40 vs. 34	97 vs. 40	55 vs. 256	25 vs. 256	42 vs. 256	15 vs. 256	24 vs. 256	16 vs. 256	12 vs. 256
Kruskal–Wallis ANOVA on Ranks (Bonferroni Correction for Pairwise Multiple Comparisons) with Asymptotic Significance at *p*-Value ≤ 0.05	*H. influenzae* vs. Healthy	*S. pneumoniae* vs. Healthy	Rhinovirus vs. Healthy	*H. influenzae* vs. Rhinovirus	*S. pneumoniae* vs. Rhinovirus	*H. influenzae* vs. *S. pneumoniae*	*H. influenzae* asymptomatic vs. Healthy	*S. pneumoniae* asymptomatic vs. Healthy	*H. influenzae* symptomatic vs. Healthy	*S. pneumoniae* symptomatic vs. Healthy	*H. influenzae* + *S. pneumoniae* vs. Healthy	*H. influenzae* + Rhinovirus vs. Healthy	*S. pneumoniae* + Rhinovirus vs. Healthy
Acetone	**0.009**	**0.013**	>0.05	>0.05	>0.05	>0.05	**0.016**	**0.027**	>0.05	>0.05	>0.05	>0.05	>0.05
Acetic Acid	>0.05	**0.040**	**0.029**	>0.05	>0.05	>0.05	>0.05	>0.05	>0.05	>0.05	>0.05	>0.05	>0.05
Dimethyl sulfide	>0.05	>0.05	>0.05	>0.05	>0.05	>0.05	>0.05	**0.016**	**0.028**	>0.05	>0.05	>0.05	>0.05
Pentanal	>0.05	>0.05	>0.05	>0.05	>0.05	>0.05	>0.05	**0.002**	>0.05	>0.05	>0.05	>0.05	>0.05
Limonene	>0.05	>0.05	**0.016**	>0.05	>0.05	>0.05	>0.05	>0.05	>0.05	>0.05	>0.05	>0.05	>0.05

A Kruskal–Wallis-Test ANOVA on ranks with Bonferroni correction for pairwise multiple comparisons for independent samples was performed between the groups and subgroups. Statistically significant differences in substance concentrations (*p*-value ≤ 0.05) are marked in bold.

**Table 3 antioxidants-13-00172-t003:** Coefficients of variation in different study groups.

	Healthy (*n* = 256)	All *H. influenzae* (*n* = 97)	Asymptomatic (*n* = 55)	Symptomatic (*n* = 42)	All *S. pneumoniae* (*n* = 40)	Asymptomatic (*n* = 25)	Symptomatic (*n* = 15)	All Rhinovirus (*n* = 34)	*H. influenzae +* *S. pneumoniae* (*n* = 24)	*H. influenzae* + Rhinovirus (*n* = 16)	*S. pneumoniae* + Rhinovirus (*n* = 12)
VOCs	RSDs (%)										
Acetone	57.4	81.7	59.7	95.3	76.9	41.5	97.2	72.0	41.5	100	69.9
Acetic acid	62.6	63.6	72.8	50.9	72.3	85.7	46.9	50.9	59.8	76.2	25.9
Dimethyl sulfide	94.8	95.0	82.4	116	83.7	66.9	112	103	118	75.6	106
Pentanal	59.8	57.5	47.9	70.8	53.3	48.9	53.9	61.9	59.6	68.3	46.9
Limonene	164	110	85.2	134	120	87.7	143	279	81.3	56.9	255

Relative standard deviations (RSDs) were calculated group-wise by rating standard deviation over sample mean and are presented in %.

**Table 4 antioxidants-13-00172-t004:** Inter-VOC correlations in different study groups.

Correlation Matrix (Dimension Reduction via Factor Analysis of Principal Components)			Healthy (*n* = 256)	All *H. influenzae* (*n* = 97)	Asymptomatic (*n* = 55)	Symptomatic (*n* = 42)	All *S. pneumoniae* (*n* = 40)	Asymptomatic (*n* = 25)	Symptomatic (*n* = 15)	All Rhinovirus (*n* = 34)	*H. influenzae +* *S. pneumoniae* (*n* = 24)	*H. influenzae* + Rhinovirus (*n* = 16)	*S. pneumoniae* + Rhinovirus (*n* = 12)
VOCs													
Acetone	Acetic acid	R-value	0.005	0.212	0.044	0.415	0.028	−0.134	0.191	0.363	0.163	**0.831**	0.134
		*p*-value	0.466	0.019	0.376	0.003	0.433	0.262	0.247	0.018	0.223	0.000	0.339
Acetone	Dimethyl sulfide	R-value	0.293	0.138	0.147	0.190	0.225	0.309	**0.527**	**0.507**	0.139	0.283	0.137
		*p*-value	0.000	0.088	0.142	0.114	0.082	0.067	0.022	0.001	0.258	0.144	0.336
Acetone	Pentanal	R-value	0.282	0.421	0.473	0.453	0.289	0.279	**0.637**	0.472	0.267	0.305	0.112
		*p*-value	0.000	0.000	0.000	0.001	0.035	0.088	0.005	0.002	0.104	0.125	0.365
Acetone	Limonene	R-value	−0.003	−0.024	−0.069	−0.011	0.256	−0.265	0.365	−0.009	0.008	0.140	−0.047
		*p*-value	0.480	0.409	0.308	0.474	0.055	0.100	0.091	0.481	0.485	0.302	0.442
Acetic acid	Dimethyl sulfide	R-value	−0.028	−0.124	−0.206	0.028	0.124	0.227	−0.172	−0.152	−0.096	0.368	**0.629**
		*p*-value	0.329	0.114	0.066	0.431	0.223	0.138	0.27	0.195	0.328	0.081	0.014
Acetic acid	Pentanal	R-value	0.198	0.296	0.267	0.381	0.257	0.323	0.122	0.166	0.065	**0.566**	**0.583**
		*p*-value	0.001	0.002	0.024	0.006	0.055	0.058	0.332	0.175	0.381	0.011	0.023
Acetic acid	Limonene	R-value	−0.005	−0.008	0.062	−0.088	0.186	0.293	0.137	−0.189	0.000	−0.114	0.474
		*p*-value	0.470	0.470	0.326	0.291	0.125	0.078	0.313	0.142	0.500	0.337	0.060
Dimethyl sulfide	Pentanal	R-value	**0.621**	0.483	0.321	**0.632**	**0.590**	**0.560**	0.425	**0.582**	0.459	**0.776**	**0.675**
		*p*-value	0.000	0.000	0.008	0.000	0.000	0.002	0.057	0.000	0.012	0.000	0.008
Dimethyl sulfide	Limonene	R-value	0.110	−0.028	−0.082	0.019	−0.031	−0.172	0.233	0.100	−0.122	0.189	**0.766**
		*p*-value	0.040	0.394	0.277	0.452	0.424	0.206	0.202	0.286	0.285	0.242	0.002
Pentanal	Limonene	R-value	0.161	−0.039	−0.087	−0.004	0.265	0.253	**0.506**	−0.027	0.095	−0.129	0.351
		*p*-value	0.005	0.354	0.265	0.490	0.049	0.111	0.027	0.439	0.329	0.316	0.132

A dimension reduction factor analysis (factor extraction via principal components method, factor score via regression method) was performed in each study group and subgroup to determine correlations between differentially expressed endogenous VOCs. Statistically significant regressions (1-tailed significance at *p*-value ≤ 0.05, R-value ≥ 0.500) are marked in bold.

## Data Availability

All data are presented within the manuscript and in the Appendix A. Any additional data or information on the study will be made available to others after reasonable request made to the corresponding author.

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
