# Peer review of "Effects of Contagious Respiratory Pathogens on Breath Biomarkers"

_antioxidants, 2024, doi:10.3390/antiox13020172_

Round 1

Reviewer 1 Report

Comments and Suggestions for Authors

The current manuscript presents a study that explores the impact of respiratory pathogens on exhaled VOCs and their potential as indicators for infection. The research utilized data from individuals at a COVID-19 test center and focused on non-SARS-CoV-2 respiratory pathogens.

Here are some major concerns and areas for modification before considering the manuscript for publication:

  1. - The study quantified 44 VOCs, excluding those from exogenous sources or below the detection limit. The quantification methods and the criteria for excluding certain VOCs need further elaboration for reproducibility and validation of the methodology​.

  2.  
  3. -Additionally, the manuscript should address the potential for confounding factors that might influence VOC levels, such as environmental factors, diet, smoke, exercize or concurrent medication use (add missing references, such as doi: 10.3390/molecules27020370, doi: 10.3390/molecules28155755 and  doi: 10.3390/bios12070520. 

  4.  

  5. -The study employed Kruskal-Wallis ANOVA for statistical comparisons between groups. However, the manuscript should provide more details on the statistical methodology, including how the researchers controlled for multiple comparisons and the potential for type I or type II errors. The selection of specific VOCs for detailed analysis should also be justified statistically​.

  6.  
  7.  

  8. -The study suggests that VOC profiling could potentially be used for non-invasive detection of pathogens. However, the manuscript should discuss the clinical relevance of these findings in more detail, considering the variability and lack of unique biomarkers for specific pathogens. The implications for clinical practice and future research directions should be clearly outlined​.

  9.  

  10. - The manuscript hints at the potential for understanding host-microbiome-pathogen interactions. This aspect could be expanded upon, discussing how the findings might influence future research in the field of respiratory infections and potential therapeutic interventions​.

- I found several minor english errors throughout the manuscript. Please have a deep language revision. 

Comments on the Quality of English Language

Moderate revision needed. 

Author Response

Our response is provided as a PDF document. 

Reviewer 2 Report

Comments and Suggestions for Authors

The paper: "Effects of contagious respiratory pathogens on breath biomarkers" by Kemnitz N. et al deals with a very interesting subject of biomarkers of the respiratory tract infections and their possible role in the distinguishing infection from colonisation. Authors critically summarize their results and demonstrate that not a single marker of infection in the exhaled breath cold have been identified. Therefore the ongoing quest for an ideal biomarker remains to be continued.

The paper is well designed, the methods are adequqtely described and the conclusions are supported by properly described results. I would recommend the paper for publication

Author Response

(The authors gave the same response as above.)

Reviewer 3 Report

Comments and Suggestions for Authors

L. 153-161:  Why should all positive subgroups and healthy be compared to each other? It is reasonable to compare each positive subgroup with the healthy group only.

L. 153-161; Figure 1: Rhinoviruses are included in “All mono-pathogens”, but then are included neither in “Asymptomatic mono-pathogens” nor in “Symptomatic mono-pathogens”. Please, clarify the reason.

Author Response

(The authors gave the same response as above.)

Reviewer 4 Report

Comments and Suggestions for Authors

please explain the type of study whether observational, retrospective or other

Please better explain the biological samples on which the tests were performed.

The inclusion and exclusion criteria should be clarified and reported

Please provide additional information about comorbidities such as COPD, pack-years

The statistical methods should be reported in a separate section

Please provide further information about symptoms and parameters of patients. It is not clear what statistical queries mean

I think it would be interesting to apply in addition to the kruskal test the correlation and regression test by relating biological markers with comorbidities and  clinical parameters

I suggest to include the following reference to improve the discussion

Expert Rev Mol Diagn. 2021 Jun;21(6):547-562. 

Author Response

(The authors gave the same response as above.)

Round 2

Reviewer 1 Report

Comments and Suggestions for Authors

Authors satisfactorily replied to my comments. Ok for me to accept.